# The Mechanoreceptor’s Role in Plantar Skin Changes in Individuals with Diabetes Mellitus

**DOI:** 10.3390/jcm10122537

**Published:** 2021-06-08

**Authors:** Claudio Zippenfennig, Tina J. Drechsel, Renan L. Monteiro, Isabel C. N. Sacco, Thomas L. Milani

**Affiliations:** 1Department of Human Locomotion, Faculty of Behavioral and Social Sciences, Institute of Human Movement Science and Health, Chemnitz University of Technology, 09107 Chemnitz, Germany; tina.drechsel@hsw.tu-chemnitz.de (T.J.D.); thomas.milani@hsw.tu-chemnitz.de (T.L.M.); 2Department of Physical Therapy, Speech and Occupational Therapy, Faculdade de Medicina, Universidade de São Paulo, São Paulo 05360-160, Brazil; renanlm@usp.br (R.L.M.); icnsacco@usp.br (I.C.N.S.)

**Keywords:** skin hardness, skin thickness, vibration perception threshold, mechanoreceptors, sensory perception, diabetic neuropathy

## Abstract

Mechanical skin properties (MSPs) and vibration perception thresholds (VPTs) show no relationship in healthy subjects. Similar results were expected when comparing MSP and VPT in individuals with diabetes mellitus (DM) and with diabetic (peripheral-)neuropathy (DPN). A healthy control group (33 CG), 20 DM and 13 DPN participated in this cross-sectional study. DM and DPN were classified by using a fuzzy decision support system. VPTs (in µm) were measured with a modified vibration exciter at two different frequencies (30 and 200 Hz) and locations (heel, first metatarsal head). Skin hardness (durometer readings) and thickness (ultrasound) were measured at the same locations. DPN showed the highest VPTs compared to DM and CG at both frequencies and locations. Skin was harder in DPN compared to CG (heel). No differences were observed in skin thickness. VPTs at 30 and 200 Hz correlated negatively with skin hardness for DPN and with skin thickness for DM, respectively. This means, the harder or thicker the skin, the better the perception of 30 or 200 Hz vibrations. Changes in MSP may compensate the loss of sensitivity up to a certain progression of the disease. However, the influence seems rather small when considering other parameters, such as age.

## 1. Introduction

Diabetes is a metabolic disease characterized by raised blood glucose levels. Hyperglycemia can lead to various secondary diseases. One of these secondary diseases is diabetic neuropathy [1]. Due to the associated loss of protective sensation, early detection of sensory nerve impairment is of great importance [2,3]. The determination of vibration perception thresholds (VPTs) can represent an important factor in the early diagnosis of diabetic neuropathy [4]. The development of diabetes leads to changes in the biomechanical properties of the skin. These can be further altered by degradation processes in the context of diabetic neuropathy [5]. Therefore, the quantification of mechanical skin properties is also an important step for ulcer risk classification in people with diabetic neuropathy [5].

The human skin consists of three layers of tissues with different mechanical and physiological characteristics. Compound keratinized squamous epithelium (keratinocytes) form the outermost layer of the skin, the so-called epidermis [6]. Beneath the epidermis is a construct of both loosely organized connective tissue (stratum papillare) and dense, irregular connective tissue (stratum reticulare)—the dermis [6]. Dermal papillae and epidermal ridges form the histologically conspicuous boundary between the epidermis and dermis, and prevent slippage of the epidermis over the dermis [6]. The third layer is the hypodermis. It generally consists of adipose tissue and thus forms the subcutaneous fat tissue, which acts as an energy reservoir and serves as thermal insulation [6]. Keratinization is the process of cell differentiation, in which the keratinocytes differentiate structurally and functionally, starting from the stratum basale up to the stratum corneum [7]. Increased and frictional loading, especially on the sole of the human foot, causes the formation of calluses, an adaptive, thickening reaction of the keratinized layer of the epidermis [8].

The human skin has a variety of different functions, such as protection, water regulation, vitamin D synthesis, thermoregulation, nonverbal communication and the sense of touch [6]. According to Jablonski, the sense of touch is the oldest sense, often referred to as “the mother of the senses” [9]. A large number of receptors, so-called low-threshold mechanoreceptors (LTMRs), ensure that humans can perceive a wide variety of sensations [10,11,12]. According to their functions, LTMRs can be divided into rapidly (RA) and slowly adapting (SA) type I and type II mechanoreceptors, with fast conduction velocities due to Aß sensory neurons [10,11,12]. In the glabrous skin, the mechanosensory end organs correspond to the following four types: Merkel cells (SAI), Ruffini corpuscles (SAII), Meissner corpuscles (RAI) and Pacinian corpuscles (RAII). RA LTMRs innervate Meissner (RAI) and Pacinian corpuscles (RAII), primarily specialized to mediate vibration and motion across the skin [11]. RAI are tuned to low-frequency vibrations (1–40 Hz). RA II primarily response to high-frequency vibrations within a range of 20–1500 Hz, with optimal activation around 200 Hz [11,12].

Individually, both VPT and mechanical skin properties have high clinical relevance. Decreased vibration perception can lead to limitations in balance [13] and gait [14] and increased risk of falls [15]. Being unable to feel pressure points due to receptor and nerve damage is another major risk factor for wounds in the foot in individuals with diabetes mellitus [2]. Changes in mechanical skin properties could contribute to this risk factor, leading to increased callus formation and changing pressure conditions while standing and walking [5,16,17]. This in turn increases the risk of foot ulceration. To our knowledge, there are only few studies that have investigated the relationship between mechanical skin properties and VPT. In a recently published paper, we showed from an evolutionary perspective that there is no relationship between mechanical skin properties and VPT [18]. The skin of individuals with diabetes changes differently than that of naturally barefoot or healthy people. Individuals with diabetes without complications show an increase in epidermal thickness, whereas individuals with neuropathy or foot ulceration have less skin thickness compared to healthy people [5]. Nevertheless, previous correlation analyses between VPT and diabetic skin characteristics come to contradictory results. Piaggesi et al. [19] found that harder skin in individuals with diabetes with neuropathy significantly correlates with the measured VPTs, while Chatzistergos et al. [20] found no relationship between the mechanical properties of the heel-pad and VPTs.

From an evolutionary point of view, calluses protect the sole of the foot without causing a loss of vibration sensitivity [18] (p. 262). Does this change from a pathological perspective? The aim of this study is to investigate the relationship between VPTs of rapidly adapting Meissner (RAI) and Pacinian corpuscles (RAII) with the mechanical skin properties of the plantar foot in individuals with diabetes mellitus (DM) and with diabetic (peripheral-)neuropathy (DPN). To better understand the interrelationships of the multifactorial disease, our study pursued the following objectives: (1) to compare VPTs of the plantar foot (heel and first metatarsal head (MET I) between a healthy control group (CG) and DM and DPN; (2) to quantify the relationship between skin hardness and skin thickness at the two anatomical sites in relation to CG, DM and DPN; and (3) to quantify the relationship between VPTs and mechanical skin properties in relation to CG, DM and DPN. Based on our results in Holowka et al. [18], we hypothesized that there is also no relationship between mechanical skin properties and VPT in DM and DPN.

## 2. Materials and Methods

A total of 66 subjects participated in this study, divided into 33 healthy controls (CG; 56.3 ± 9.9 years, 1.7 ± 0.1 cm, 70.1 ± 11.9 kg, 14 ♂: 19 ♀) and 33 individuals with diabetes mellitus (without (DM; n = 20, 53.3 ± 15.1 years, 1.6 ± 0.1 cm, 77.9 ± 13.3 kg, 7 ♂: 13 ♀, DM duration: 11.9 ± 10.3 years) and with diabetic (peripheral-)neuropathy (DPN; n = 13, 61.0 ± 14.5 years, 1.6 ± 0.1 cm, 81.8 ± 17.9 kg, 7 ♂: 6 ♀, DM duration: 17.5 ± 8.7 years)). A fuzzy decision supporting system was used to classify individuals with diabetes as with DPN [21,22]. All participants gave their written consent to participate. This study was performed in accordance with the recommendations of the Declaration of Helsinki and approved by the Ethics Committee of the University of São Paulo (CAAE 54283516.3.0000.0065).

Before VPT measurements, participants underwent a 10-min acclimatization period to room temperature. During this time, mechanical skin properties were determined. VPTs were measured in a prone position at two different frequencies (30 and 200 Hz) and anatomical locations (first metatarsal head (MET I) and heel) of the left or right foot. These two frequencies are considered ideal for measuring RAI and RAII [11,12]. The foot, frequency and anatomical location tested for each subject were randomized. Using a swivel arm, the probe (diameter 7.8 mm) of a modified vibration exciter (Mini-Shaker type 4180, Brüel and Kjaer Vibro GmbH, Darmstadt, Germany) was placed precisely perpendicularly at one of the anatomical locations (Figure 1a). The vibration exciter was powered by a powerbank (XTPower MP-3200, Batteries and Power Solutions GmbH, Ellwangen, Germany). The vertical movement of the vibration exciter’s probe was calibrated by using a high-precision capacitive position sensor (CS05, Mirco-Epsilon Messtechnik GmbH & Co. KG, Ortenburg, Germany). The vibration amplitude (in µm) was calculated by using an acceleration sensor (MMA2240KEG, NXP Semiconductors Netherlands B.V., Eindhoven, Netherlands). An integrated force sensor (DS050A9, disynet GmbH, Brüggen-Bracht, Germany) was used to adjust the force the probe exerted against the skin (intended range 1.0 ± 0.2 N). VPTs were measured three times for each anatomical location, using a customized vibration threshold protocol based on previous studies [18,23,24]. Participants had to press a button as soon as they felt a vibration. The algorithm started with an above-threshold sinusoidal vibration burst (2 s duration), which was perceived by the subject. If the subject does not feel the starting amplitude, then our algorithm cannot start and the measurement of the VPT cannot take place. In this study, each participant was able to feel the starting amplitude. The vibration amplitude was then halved until the subject did not feel the burst for the first time. Then, the average amplitude from the last perceived and the last undetected burst was tested. The protocol ended four bursts after the first undetected stimulus. Our algorithm randomly varies the pause time between two consecutive vibration stimuli (2–7 s). Thus, we are trying to prevent anticipation and the subject cannot get used to a rhythm. The mean of the smallest perceived and largest unperceived vibration amplitude was recorded as VPT. The mean out of three VPT measurements was used for statistical analysis.

Mechanical skin properties were measured at the same anatomical locations as those used to determine VPTs. A Shore OO Durometer (AD-100, Checkline Europe GmbH & Co. KG, Bad Bentheim, Germany) was used to measure skin hardness. Subjects were asked to flex their knees to approximately 90 degrees so that the sole of the foot pointed horizontally upwards. The probe (diameter 2.4 mm) was then applied perpendicularly to the anatomical locations (Figure 1b). Indentation depth (Shore OO hardness units (Sh)) can be read by using an analogue scale, where 0 is the softest and 100 is the hardest. Therefore, hardness is defined as the indentation depth created by a defined pressure [18]. The mean out of three measurements was used for statistical analysis.

Skin thickness was determined by using a handheld ultrasound device (L7, Clarius Mobile Health Corp., Burnaby, Canada). The transmission frequency of the scanner was optimized to 10 MHz for imaging and automatically adjusts according to the scanning depth. Skin thickness was measured with subjects in prone position. Ultrasound gel was applied to the corresponding anatomical locations, and three ultrasound images were captured for each site. The sharpest of these three images was evaluated by two independent investigators using ImageJ software. The distance (in mm) was measured between the two superficial hyperechoic lines representing the borders of the epidermis [25] (Figure 1c). The mean of the two thickness measurements was used for statistical analysis.

Differences in anthropometric data between groups were examined by using Kruskal–Wallis rank sum tests or ANOVAs, depending on data distribution, followed by Bonferroni post hoc corrections. DM duration was compared between DM and DPN groups, using the Wilcoxon rank-sum test for independent samples. VPTs were log-transformed to achieve normality and correct the naturally skewed distribution for statistical analysis [26]. T-tests for dependent samples and Wilcoxon signed-rank tests were performed to compare the VPTs at the two frequencies and anatomical structures, and to analyze differences between the mechanical skin properties at the two measurement locations. Kruskal–Wallis rank sum tests and ANOVAs were performed to analyze differences between groups. Post hoc tests for pairwise comparisons were performed with appropriate Bonferroni corrections. Spearman’s rank-order and Pearson’s product-moment correlation coefficients were used to test for relationships within the mechanical skin properties and between VPTs and mechanical skin properties. General linear models were used to test the relationship between VPTs and skin thickness, with age and skin hardness set as covariates, and gender and group (CG, DM and DPN) set as fixed effects.

## 3. Results

There were no significant differences for age (*p* = 0.241) and bodyweight (*p* = 0.055) between groups. The ANOVA for height was statistically significant (*p* = 0.049). However, there were no significant post hoc comparisons. The differences in diabetes duration were not statistically significant (*p* = 0.070).

VPTs at 30 Hz were significantly higher compared to 200 Hz at both locations for all groups (all *p* < 0.01). While VPTs at 200 Hz show no differences between the two measurements sites, VPTs at 30 Hz at the heel of the CG (*p* < 0.01) and the DM (*p* < 0.01) were significantly higher than at MET I. Interestingly, there was no significant difference between the two measurement sites for 30 Hz VPTs in DPN. Furthermore, DPN VPTs were significantly higher (all *p* < 0.01) compared to DM and CG at both frequencies and locations. Surprisingly, we found no difference between DM and CG (Table 1).

There were no significant differences between measured anatomical structures in all three groups with regard to skin thickness. While there was also no significant difference between the measurement sites in the skin hardness of CG, the skin in the heel area in DM (*p* = 0.019) and DPN (*p* = 0.013) was significantly harder than at the MET I. Skin hardness at the heel was significantly higher for DPN compared to CG (*p* < 0.01). There were no significant differences for skin thickness between groups (Table 1). Considering all groups together, skin hardness and thickness correlated significantly at the heel (*p* < 0.01, rs = 0.45), but not at MET I (*p* = 0.13, rs = 0.23). When looking at the groups individually, moderate-to-not-significant positive correlations were found for CG (Figure 2). While DM showed a moderate correlation especially at MET I, DPN showed a strong correlation at the heel (Figure 2).

Furthermore, 30 Hz VPTs at the heel and MET I showed moderate-to-high negative correlations (*p* = 0.020 and *p* = 0.187, respectively) with skin hardness for DPN (Figure 3a,b). For 200 Hz VPTs at the heel and MET I, moderate negative correlations with skin thickness (*p* = 0.181 and *p* = 0.120, respectively) were found for DM (Figure 3c,d). Additionally, 200 Hz VPTs at the heel showed a moderately positive correlation with skin thickness for DPN (Figure 3c). No other correlations were found.

The general linear models found no effects of mechanical skin properties on VPT at either location or frequency. The significance of model effects was tested by using type-3 ANOVAs on model variance.

## 4. Discussion

From an evolutionary stand point, a callus protects the sole of the foot without leading to a loss of vibration sensitivity [18]. Therefore, the present study investigated this from a pathological point of view. It is known that diabetes leads to changes in vibration sensitivity [4]. In the present study, the VPTs of the DPN were consistently higher at both frequencies and anatomical locations compared to the CG and DM, implying that these subjects were more insensitive. Interestingly, there were no differences between the VPTs of the CG and DM (Table 1). This aspect is discussed in detail elsewhere [27]. The focus of this study is on the relationship between mechanical skin properties and VPTs in individuals with diabetes mellitus. Based on the study by Holowka et al. [18], we hypothesized that there is no relationship between the mechanical skin properties and VPT in individuals with and without DPN.

Interestingly, in contrast to Chao et al. [5], there were no significant differences in skin thickness between the heel and MET I for all three groups. The reason could be the considerably different transmission frequencies of 55 MHz in Chao et al. [5] versus 10 MHz in the present study. The higher the resolution of the ultrasound system, the more accurately the epidermal thickness can be visualized [28]. Unfortunately, we were limited to the technical capabilities of the equipment. Nevertheless, descriptively, our results are consistent with those of Chao et al. [5]: the epidermis of the heel was thicker than at MET I in all groups, and the epidermal thickness at DM was increased compared to CG but decreased again as disease severity progressed (DPN).

For skin hardness, our results are mostly in line with Piaggesi et al. [19] and Periyasamy et al. [29], who found that skin hardness increases in individuals with diabetes mellitus and with neuropathy. However, it is interesting to note that in the present study, the heel was significantly harder than MET I in both DM and DPN (Table 1). The data of Periyasamy et al. [29] show the opposite result, which could be attributed to a different methodology for measuring skin hardness. As suggest by Falanga et al. [30], in the present study, the Durometer was applied perpendicularly to the skin and only by its own weight. For this purpose, the subjects laid in prone position and bent their knees approximately 90 degrees so that the sole of the foot pointed horizontally upwards (Figure 1b). In the study by Periyasamy et al. [29], the subjects were lying supine with their feet held vertically and the tips of their toes pointed upwards. The probe was then pressed against the skin. The methodology of measuring skin hardness can be compared to Piaggesi et al. [19]. Unfortunately, Piaggesi et al. [19] did not measure areas with bony prominences so that a comparison to our measurements at MET I is not possible. The skin hardness in DM and DPN should be further investigated, possibly in a kind of mapping at different plantar sites.

The relationship between skin thickness and skin hardness is somewhat different than in previous studies with healthy subjects. Strzalkowski et al. [25] and Holowka et al. [18] found strong positive correlations between skin thickness and skin hardness in healthy subjects. In our CG, moderate to no significant positive correlations were found between these two parameters. The reason for this could be the large difference in the age of the subjects in both studies. The comparison group in Holowka et al. [18] (usually shod group) had a mean age of 35 years, while the subjects of Strzalkowski et al. [25] had a mean age of 24 years. The mean age of the CG in this study was 56 years. The skin shows age-related changes by becoming more lax and thinner [28,31]. In addition, it has already been demonstrated that older subjects over 35 years of age have thinner skin than younger subjects [28]. These age-related skin changes could be the cause of the different correlation result in CG. We found moderate to strong significant correlations between the two skin parameters. However, only for the heel in DM and MET I in DPN (Figure 2). The difference in skin hardness between DM and DPN may be the result of higher pressures and loads in the heel area during walking, caused by the sensitivity loss in the DPN group [16]. To our knowledge, to date, no studies have investigated the relationship between skin thickness and skin hardness in DM and DPN. Partially blurred ultrasound images thinned out the already small samples of DM and DPN (12/20 and 7/13, respectively). Thus, further investigations are necessary to generate possible explanatory approaches.

In contrast to Chatzistergos et al. [20], the mechanical skin properties of individuals with diabetes in our study influenced their VPTs. Similar to the results of Piaggesi et al. [19], DPN had the hardest skin compared to DM and CG (Table 1). Interestingly, Piaggesi et al. [19] found a positive correlation between skin hardness and VPT. Unfortunately, the measurements of skin hardness and VPTs in Piggesi et al. [19] did not take place at the same anatomical locations as in the present study. Our study measured skin hardness and VPT at the same anatomical locations, resulting in moderate to high negative correlations with skin hardness for VPTs at 30 Hz. This means, the harder the skin, the better the perception (Figure 3a,b). In DPN, earlier epidermal denervation compared to deeper dermal layers could result in structural changes and might explain the correlations found [3]. Consequently, sensitivity loss in superficial RAI may occur earlier and greater than in RAII in deeper tissues. Thus, RAII and their pathways seem to be influenced less than RAI. Hardening the skin may be the body’s attempt to compensate for this loss of sensitivity. Recent studies showed that increasing contact force and/or stimulation area improved VPTs even at lower frequencies [32,33]. Hardening the skin could lead to a wider spread of vibrations, stimulating a higher quantity of remaining mechanoreceptors and their afferences.

Furthermore, we observed moderate negative but not significant correlations between 200 Hz VPTs and skin thickness for DM (Figure 3c,d), which showed the thickest skin compared to DPN and CG. Again, the thicker the skin in DM, the better the perception. At the onset of diabetes, RAI show hypertrophic and structural changes, while RAII only show structural changes [34]. Furthermore, fibrous collagen networks show stronger crosslinking [35], leading to skin thickening [5]. The Durometer measured the superficial stiffness of the skin. Deeper plantar stiffness could not be measured, but possibly was quantified indirectly via skin thickness measurements. Skin thickness is directly related to skin stiffness due to the accumulation of glycosylation end products [5,35]. An increased spatial summation of RAII due to the increased plantar stiffness could therefore provide an explanation for the enhancement of high-frequency VPTs in DM [36]. To confirm this theory, further studies measuring the thickness and stiffness of the total plantar soft tissue, similar to Chao et al. [5], are necessary. Furthermore, our theory is based on changes that occur in the early stages of diabetes [34]. Our range in diabetes duration in DM is high (11.9 ± 10.3 years), which is why further studies should compare subjects newly diagnosed with diabetes with subjects with long-term diabetes.

Besides mechanical skin properties, VPTs may be influenced by other factors, such as age or gender. To quantify the influence of mechanical skin properties in relation to these parameters, we calculated different general linear models. Similar to previous findings [18], only a significant influence of age and group were found for all VPT conditions, and an influence of gender was only found under 30 Hz conditions. Studies have shown that older subjects are less sensitive to VPT, and that men are less sensitive at lower VPT frequencies than women [37]. From the age of 50, men have higher VPTs than women, because of the assumed faster degeneration of the peripheral nervous system [37]. This gender effect was only measured at 30 Hz, supporting the integration of low vibration frequencies for the diagnosis of diabetes-associated changes [4,38]. Consequently, men with diabetes are even more affected than women [39] and healthy subjects. Furthermore, DPN consistently had higher VPTs [40] at both frequencies and anatomical locations. This is consistent with the results of the general linear models.

From an evolutionary point of view, calluses protect the sole of the foot without causing a loss of vibration sensitivity [18] (p. 262). From a pathological perspective, in individuals with diabetes, the changes in mechanical skin properties may be a compensatory mechanism in response to the loss of sensitivity up to a certain progression of the disease. The positive correlation between skin thickness and 200 Hz VPT in DPN (Figure 3c) may indicate a reversal point of compensation: tissue disappears, thickness decreases and hardness increases. These changes could affect the functionality of RAII by affecting the compression of their lamellar structure [34]. Nevertheless, these changes do not have as great an influence on sensory perception as other parameters (e.g., age), but should be considered in further studies with larger samples.

Our results are relevant from a clinical perspective regarding two main aspects: (1) As soon as patients are newly diagnosed with diabetes, the first medical meetings focus, among other things, on the relevance to self-monitoring of the feet. This is mainly about recognizing unnoticed injuries as early as possible. Additionally, in the sense of an even stronger education, the patients should focus on existing skin changes, e.g., callus formation. Based on our data, skin changes could be an indication of incipient sensory changes. (2) Furthermore, the aim of DPN diagnostics is to detect sensorial and musculoskeletal changes in the multifactorial disease as early as possible. VPTs, especially at low-frequency vibrations, seem to be suitable for this purpose [4,38]. Recording mechanical skin properties can also contribute to an earlier detection of DPN changes [19]. Both aspects, but especially the clinical observation of changing skin properties, should be intensified in DM diagnostics. The durometer in particular is an easy-to-use and quick measuring device that could expand clinical diagnostics and be used in the context of podiatric treatments. In addition, to our knowledge, there are currently no studies examining the relationship between podiatric treatments and possible sensory changes. In our study, we did not record the status of medicinal or podiatric treatments. For this reason, future studies should investigate the influence of medicinal, skin lotions and podiatric treatments on VPTs and mechanical skin properties in individuals with DM.

## 5. Conclusions

The harder the skin in DPN, the better the perception at 30 Hz vibrations. The thicker the skin in DM, the better the perception at 200 Hz vibrations. From a sensory perspective, skin changes associated with DM could compensate for the onset of sensory loss up to a certain point of the disease. Nevertheless, taking into account other DM influencing factors (e.g., age), the influence of mechanical skin properties on sensory perception seems to be small.

## Figures and Tables

**Figure 1 jcm-10-02537-f001:**
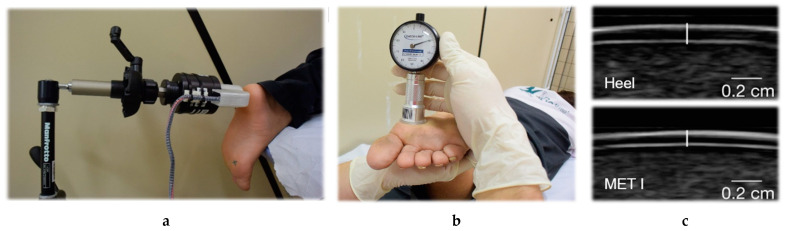
Measurement setup for quantifying the relationship between vibration sensitivity and mechanical skin properties. (**a**) Measuring vibration perception thresholds with the customized vibration exciter. (**b**) Measuring skin hardness with a handheld Durometer. (**c**) Quantifying skin thickness with a handheld ultrasound transducer at the heel and first metatarsal head (MET I) [18].

**Figure 2 jcm-10-02537-f002:**
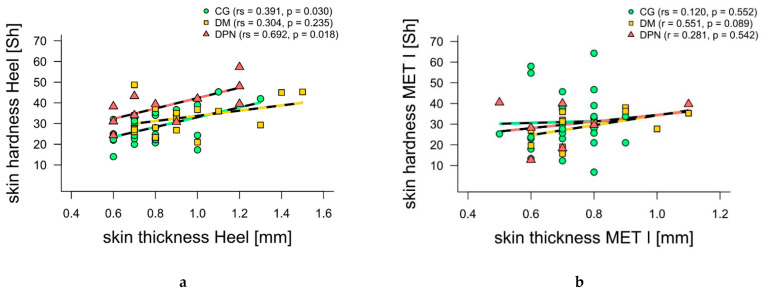
Spearman’s rank-order (rs) and Pearson’s product-moment (r) correlations between skin hardness and skin thickness separated by groups. (**a**) Scatter plot of skin hardness versus skin thickness at first metatarsal head (MET I) for control group (CG, green), individuals with diabetes without diabetic neuropathy (DM, yellow) and individuals with diabetes with diabetic neuropathy (DPN, red). (**b**) Scatter plot of skin hardness versus skin thickness at heel for control group (CG, blue), individuals with diabetes without diabetic neuropathy (DM, yellow) and individuals with diabetes with diabetic neuropathy (DPN, red).

**Figure 3 jcm-10-02537-f003:**
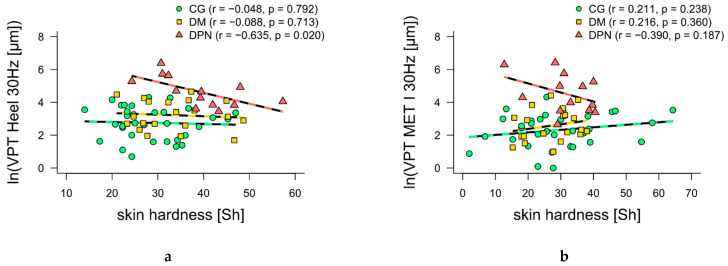
Spearman’s rank-order (rs) and Pearson’s product-moment (r) correlations between vibration perception thresholds (VPT) and mechanical skin properties. (**a**,**b**) Scatter plot of skin hardness versus VPT at 30 Hz at the heel and first metatarsal head (MET I) for control group (CG, green), individuals with diabetes without diabetic neuropathy (DM, yellow) and individuals with diabetes with diabetic neuropathy (DPN, red). (**c**,**d**) Scatter plots of skin hardness versus VPT at 200 Hz at heel and first metatarsal head (MET I) for control group (CG, green), individuals with diabetes without diabetic neuropathy (DM, yellow) and individuals with diabetes with diabetic neuropathy (DPN, red).

**Table 1 jcm-10-02537-t001:** Vibration perception thresholds and mechanical skin properties in healthy subjects and individuals with diabetes.

	Variable	CG	DM	DPN	Statistics	d.f.	*p*-Value
Heel	VPT 30 Hz (µm)	15.4 [7.3–31.3] ^#^	23.0 [14.4–57.9] ^§^	104.4 [46.5–196.9] ^#§^	18.72	2	<0.01
	VPT 200 Hz (µm)	2.0 [0.6–5.9] ^#^	1.1 [0.7–4.4] ^§^	18.0 [5.1–31.0] ^#§^	9.52	2	<0.01
	thickness (mm)	0.78 ± 0.17	0.94 ± 0.41	0.86 ± 0.40	5.18	2	0.08
	hardness (Shore 00)	28.6 ± 7.9 *	33.4 ± 8.3	39.0 ± 8.8 *	7.89	2	<0.01
MET I	VPT 30 Hz (µm)	10.7 [4.9–19.9] ^#^	9.9 [6.9–24.1] ^§^	73.9 [33.6–191.8] ^#§^	21.29	2	6 × 10^−8^
	VPT 200 Hz (µm)	1.9 [0.4–8.6] ^#^	1.1 [0.5–4.7] ^§^	23.7 [16.5–51.1] ^#§^	12.82	2	<0.01
	thickness (mm)	0.72 ± 0.10	0.78 ± 0.41	0.71 ± 0.40	1.56	2	0.46
	hardness (Shore 00)	29.1 ± 14.0	27.1 ± 7.5	31.3 ± 8.4	0.56	2	0.57

Notes: *p*-values, statistics and degrees of freedom (d.f.) are from Kruskal–Wallis rank sum tests for skin thickness and ANOVAs performed for skin hardness and VPTs. Pairwise comparisons using t-tests with pooled SD were done for all VPTs (^#, §^ all *p*-values < 0.01) and for skin hardness at the heel (* *p* < 0.01). *p*-values were adjusted by using the Bonferroni method. VPT values are reported as median and interquartile range. Mechanical skin properties are reported as mean ± SD.

## Data Availability

The data presented in this study are available on request from the corresponding author. The data are not publicly available due to further analyses.

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
