# Peer review of "The Mechanoreceptor’s Role in Plantar Skin Changes in Individuals with Diabetes Mellitus"

_jcm, 2021, doi:10.3390/jcm10122537_

Round 1
Reviewer 1 Report
The paper investigates vibration perception threshold in two different frequencies and skin hardness and thickness in three different groups: 1) Non-diabetic control subjects, and individuals with or without diabetic neuropathy. Although the size of each group is very small the paper addresses interesting clinical questions.
First some general comments: The authors should consider using less abbreviations which would make the paper more easily accessible. For example, why not write “heel” instead of using letter H?
Is there any more data on diabetic subjects? Age, duration, sex, and height in both patients with diabetic neuropathy and those without? This is important since it may affect the VPTs and explain some of the results.
As the authors point out VPTs are not normally distributed, and the data should therefore be presented as median and interquartile range. Presenting means and standard deviations results in strange table where the lower confidence interval of the VPT in DPN group at 0 Hz is a negative value!
At line 148 you write that “VPTs at 30 Hz were significantly higher compared to 200 Hz at both locations for all 148 groups (all p < 0.01).” This is certainly true for the amplitude but since the Acceleration A = [2π f]2 D where D is peak displacement and f=frequency) . (These values can be given as Decibel (dB); see for example Ekman et al. https://pubmed.ncbi.nlm.nih.gov/33822804/ for reference values for different frequencies at different location and different age groups) and the energy needed to stimulate so that the person can perceive vibrations is more at high frequencies than at low frequencies.
A question related to the previous one: Did the instrument have a ceiling effect and if not, how did you manage to avoid noise at 200 Hz frequency? If the answer is yes, how many of the patients in DPN group reached the upper limit of the VPT without being able to feel it?
“The mean of the 111 smallest perceived and largest unperceived vibration amplitude was recorded as VPT. “ – Why not use the mean of all 6 points? Would it change the results?
In our experience there are persons who cannot perform this kind of measurement and try to quess the results which can be noticed if several frequencies are tested. Did you exclude any subjects because of obviously trying to guess?
“Interestingly, there was no significant difference between the two measurement sites for 30 Hz VPTs in DPN.” Could it be because of the very small size of the DPN group?
Does any medication affect skin hardness? Skin lotions that we recommend the patients with dry skin (often DPN) to use?
Results:
It seems that individuals with diabetes had lower VPTs than persons without. It seems however somewhat contradictory that increased plantar stiffness due to accumulation of (perhaps you mean advanced glycation endproducts? ) leads to better VPTs . Has this been shown previously? Most of the studies show that individuals with diabetes have higher VPTs than non- diabetic subject or if similar if duration is short. Could other differences like age, height or sex explain this result?
Line 283: that afferent degeneration in individuals with diabetes can already be 283 detected at 30 Hz rather than at 200 Hz [5,34]. Reference number 5 suggests the opposite which mean that earliest changes can be seen at higher frequencies.
Reviewer 2 Report
The work titled “The Mechanoreceptor’s Role in Plantar Skin Changes in Individuals with Diabetes Mellitus” addresses a topic of interest to be published in JCM. The work consists of a control group and two study groups that can with a small number of subjects that can cause the results to be questioned. Despite this, there are a number of methodological issues that should be addressed by authors in order to accept the work for publication.
The summary must be reformulated, it does not reflect the parts of the work, specifically the methodology, results and conclusions. In addition, the authors make use of abbreviations that were not previously explained.
The introduction is well written and follows a reading thread, but I consider that it was developed too much on the skin, functions and sensitivity.
The work does not clearly detail the objective it pursues.
In table 1 the title and legend should be unified.
The work of finishing with the conclusions based on the results obtained in the work, which must answer clearly and directly to the objective set in the work.
I observe an absence of reflections or contributions by the authors in the discussion that they address about what this work contributes to daily clinical practice.
Round 2
Reviewer 2 Report
The authors responded cordially and took into account the comments made, which improved the quality, reading and understanding of the article. But I think that the conclusions should have its own section and try to be more specific. And I still think that the authors do not address in depth the clinical contribution of this work in the discussion section
Author Response
Point 1: The authors responded cordially and took into account the comments made, which improved the quality, reading and understanding of the article. But I think that the conclusions should have its own section and try to be more specific. And I still think that the authors do not address in depth the clinical contribution of this work in the discussion section.
Response 1: Thank you for your comment. We would like to thank you again for your comments in the first round and are pleased that you feel we have taken them into account satisfactorily. We have given our conclusions their own section and have tried to be more specific and concise. Furthermore, we have explicitly highlighted the clinical contributions of our work in the discussion.